# Does working from home work? That depends on the home

**Martijn Stroom**[ID]*, **Piet Eichholtz**[º], **Nils Kok**[ID][º]

Department of Finance, School of Business and Economics, Maastricht University, Maastricht, The Netherlands

º These authors contributed equally to this work.
* m.stroom@maastrichtuniversity.nl

**Data Availability Statement:** All files are available from the OSF database (see https://doi.org/10.17605/OSF.IO/H6J3F).

**Funding:** The author(s) received no specific funding for this work.

## Abstract

Working from home (WFH) has risen in popularity since the COVID-19 pandemic. There is an ongoing debate about the productivity implications of WFH, but the physical climate of the home office has received only limited attention. This paper investigates the effect of home office satisfaction and environment-improving behavior on productivity and burnout tendency for WFH employees. We surveyed over 1,000 Dutch WFH individuals about their home office and perceived WFH performance. We fit logistic regressions and structural equation models to investigate the effect of home office satisfaction and characteristics on self-reported productivity, burnout tendency, and willingness to continue WFH. Our results reveal that individual differences in WFH productivity are explained by heterogeneity in the physical home office environment. Higher satisfaction with home office factors is significantly associated with increased productivity and decreased burnout tendency. We continue by showing that more ventilation during working hours is associated with increased productivity, willingness to continue WFH, and burnout resilience. This effect is fully mediated by satisfaction with the home office. We find that higher home office satisfaction is associated with WFH success and air-quality-improving behavior is associated with higher satisfaction. Our results underline a holistic perspective such that investing in a healthy and objectively measured physical climate is a key aspect of the bright future of working from home. The move from the work office to the home office needs to be accompanied by careful design and investment in the quality of the office and its climate.

## Introduction

The COVID-19 pandemic, in combination with recent technological advancements, has quickly elevated the status of working from home (WFH) from "occasionally" to "the new normal" [1]. Earlier uncertainty about the quantity and quality of work produced at home had hampered large-scale corporate acceptance [2, 3]. However, these doubts were simply overturned by the COVID-19 pandemic, which forced most knowledge-based employees to work online. Negative stigmas that were previously associated with WFH diminished drastically, at least temporarily [1]. In addition, prior technological complications were quickly overcome

**Competing interests:** The authors have declared that no competing interests exist.

following a pandemic-driven surge in technological innovations, such as the advent of Teams and Zoom calls. This involuntary litmus test pushed WFH out of its infancy. However, what has gained limited attention is the physical climate of the home office in which work takes place. This study investigates the relationship between the home office environment, including available hardware (e.g. computer, chair, etc.) but also environmental conditions (e.g. air quality, temperature, etc.) and self-reported measures of work satisfaction, productivity and burn-out tendency.

## Work from home: Productivity and performance

The rising popularity of WFH has been well-reported: a recent report by buffer.com [4] among 2,300 employees showed that over 97% would like to continue to work from home, at least partially. Employees are, on average, willing to take a 5% pay cut for 2–3 days of work from home [5]. Employees working from home report being as productive as they were at the office before the pandemic [6]. These positive experiences have led to the prediction that, after the pandemic, 20% of all office work will be carried out from home. This continuation of work from home is expected to boost productivity by almost 5%, although largely unobservable by standard measures, as it stems mainly from a reduction in commuting [1].

Working from home has clear advantages, as well as disadvantages, for both work performance and human health and well-being. Multiple studies show positive effects on job satisfaction and turnover intent [7–9]. Bloom et al. [10] report that work from home leads to less commuting and fewer distractions. In addition, exhaustion leading to burnout is negatively related to work from home [11]. Perceived autonomy seems to be one of the main drivers of these positive effects: the degree to which employees can choose a location and time to work, independently of their supervisors, both predict the intensity of working from home, as well as job performance, mental burnout, and job dedication, even during the pandemic [10–13].

More recently, Bloom et al. [14] found only modest self-reported and realized productivity increases for WFH during COVID-19, whereas others identified productivity decreases for those who did not WFH before the pandemic, suggesting selection bias in previous studies [15]. Moreover, output assessments among ICT workers suggest productivity actually drops at home [16]. In the past, the positive relationship between WFH intensity and productivity has repeatedly been found to be non-linear. Golden & Vega [17] find that the relationship between WFH intensity and productivity is nonlinear, with optimal productivity at 16 WFH hours per week, beyond which job satisfaction and performance decline. A survey by State of the Work in 2022 found that, among 2,000 respondents, 45% think career growth will be at risk with increased WFH [18]. Unsurprisingly, it is coworkers' relationships that suffer most from WFH, leading to professional isolation, which in turn has the potential to escalate into decreased performance and increased turnover intent [9]. Offline or online communication could mitigate these negative effects, but only partially [13, 19]. For instance, Yang et al. [20] find that firm-wide remote work inevitably lowers communication quality, as less communication leads to a worsening of information sharing.

Beyond having implications for coworker relationships, WFH may also bring new interpersonal problems to light. Felstead & Henseke [21] suggest that homeworkers are burdened by the "social exchange theory": they work harder, longer, and work unpaid hours in order to justify their freedom to work from a preferred location. Workers thus (over)compensate for the perception that they might work less when not being observed. The resulting work exhaustion may offset the positive effects of WFH on productivity, and may even lead to burnout symptoms [22]. In addition, research shows that people working from home find it hard to detach from work, disrupting their work-life balance [13, 23]. Interestingly, the work-family conflict

was previously considered to decrease with WFH, supposedly due to increased autonomy [9]. The current perception of WFH having a negative impact on work-life balance could therefore also be a pandemic-specific challenge.

Although academic findings on the implications of WFH vary, it is also important that beyond the average effects, substantial heterogeneity has been documented across jobs and individuals. To our knowledge, this heterogeneity has solely been explained by work and personal characteristics. For instance, the degree to which a job is suitable for WFH strongly predicts productivity [6]. A job previously executed behind a desk (e.g., financial services) is more easily shifted to a home office as compared to a manual, labor-orientated occupation. A heavy workload and the degree of monitoring by supervisors also negatively impact the work effectiveness from home [13]. Jobs that have high levels of interdependence with colleagues, or are outcome-oriented, suffer when WFH intensity increases [24]. Overall, limited support and inadequate feedback by the employer mitigate the positive effects of WFH [11, 13].

At the individual level, self-discipline seems to be a key factor in explaining the effectiveness of WFH [13]. The degree to which an individual is able to ignore distractions that are not present at the office is important, especially without the same level of social control by co-workers. Additionally, women seem to suffer more from WFH as compared to men [6]. Women state their job to be less suitable for WFH in general and the presence of children affects WFH productivity for women more negatively as compared to men [25–27]. Finally, the pandemic showed that young workers seem to appreciate work from home more, and opted for WFH more often as compared to older workers [28]. These results, however, are not stable per se. Another study shows opposite results, where both women and older workers reported being more productive when WFH [29].

## Work from home: The role of the physical environment

What has gained limited attention in explaining individual differences in WFH satisfaction and productivity is the physical climate in which daily work takes place. The COVID-19 pandemic has led to increased attention to the effect of air quality in indoor spaces on pathogen spreading. Specifically, ventilation has become the spearhead combating the airborne spreading of the COVID-19 virus at public and private indoor gatherings [30, 31]. The attention to air quality reinforces an existing trend in which workplace quality is becoming more and more important. In the office, employers aim to facilitate a healthy and comfortable work environment for employees, with the goal of promoting productivity [32–34]. Suboptimal air and light quality, temperature, and noise have all been shown to negatively affect productivity and increase sick building symptoms, such as headaches, in the office [35–38]. Hence, ergonomics, temperature, and noise pollution are all considered by modern employers in order to minimize interference with comfort and wellbeing (and ultimately: productivity) in the office [39].

For the move to the home office, a trade-off is to be expected. On the one hand, suboptimal ergonomics at home are not as easily mitigated [40], and workplace professionalism or quality may suffer [41]. For instance, not having a dedicated office negatively influences productivity at home [29]. On the other hand, research suggests that controlling the thermostat at home might benefit WFH satisfaction [42, 43]. Looking at indoor environmental quality more broadly, Tahmasebi et al. [44] show that people working at home during the pandemic close their windows more often as compared to before the lockdown. Combined with $CO_2$ data, they conclude that WFH often leads to worse indoor air quality. Generally, the professionalism or quality of the work environment might suffer, while people's experienced control over these conditions at home might increase.

To address this knowledge gap, the current study examines the relationship between the home office and work-from-home success. We hypothesize that higher satisfaction with the physical environment in which WFH is being performed is associated with higher perceived productivity and lower burnout tendency. Moreover, in line with the recent research focused on air quality and performance in controlled settings, we hypothesize that improving the home office air quality through ventilation will be associated with higher office satisfaction and subsequent work-from-home outcomes.

## Method

### Survey participants

We surveyed 1,002 Dutch individuals via the Flycatcher panel. Flycatcher is an academically-oriented research organization that established a high-quality panel representing the Dutch population (for example, see [45–47] for studies using the Flycatcher panel). Flycatcher randomly selected participants from their panel for an online survey, where participation was reimbursed. All Flycatcher participants received written informed consent, were allowed to drop out at any time, and included participants actively consented to participation ('double-active-opt-in'). For the purpose of our research, we included just office workers (with a minimum age of 18 years old), who worked at least part-time from home at the time of the survey. People without work, previously without work, or working exclusively from the office were excluded from our sample. All data was collected unanimously and thus cannot be traced to an individual in the panel. The research setup was reviewed and approved by Maastricht University's Ethical Review Committee Inner City Faculties (ERCIC_195_09_06_2020).

### Empirical setting

The data collection took place in November 2020. At that time, the Netherlands had been in some form of lockdown for over 8 months due to the COVID-19 pandemic. The government strongly recommended WFH, with the exception of healthcare and other essential workers, and prevented employers from requiring employees to work in person. During this time, employers were not allowed to force their employees to come to the office, and social activities were severely limited. Respondents were asked to answer a selection of questions based on two moments in time: current (working from home) and one year ago (working from the office). Fig 1 provides an overview of the timing of data collection relative to the development of COVID-19 restrictions.

It is relevant to point out that we utilize the COVID-19 restrictions to eliminate selection problems hampering previous research. Before the pandemic-related restrictions, the success and satisfaction of WFH could potentially be explained by self-selection following the request

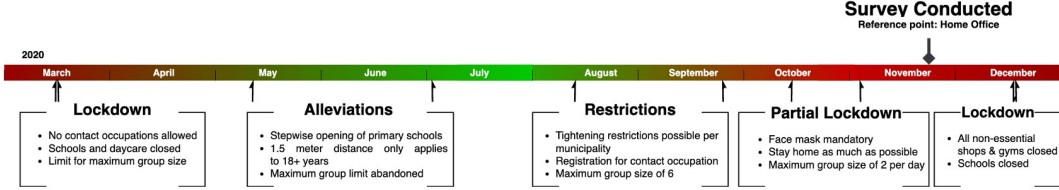

**Fig 1. COVID restrictions and survey timeline.** Timeline of Dutch national COVID-19 policies in 2020, color-coded by restriction intensity. Dark red represents the most stringent restrictions, while light green represents the most liberal policies from a social perspective. Key events include lockdowns, partial lockdowns, and periods of alleviations and restrictions, with specific measures noted at each stage.

to (voluntarily) move to work from home. Inherent intrinsic motivation, personal characteristics, and ability to adjust to the physical environment could all be omitted factors in that request. From a company perspective, those previously offered the possibility to WFH likely had job characteristics with at least a partial fit with remote work. Due to the pandemic, the susceptibility to selection bias is eliminated, leading to a clean research setting to evaluate the impact of WFH on satisfaction, productivity and burnout.

## Material and variable construction

The survey included several previously validated modules. First, in order to measure productivity and work satisfaction, the survey included the Health and Work Questionnaire (see [48–50]. Following a cluster analysis, a revised version was developed, more specifically fitting the working-from-home situation (WFH-HWQ) [51]. This easily-administered questionnaire allows for the assessment of various factors of work-related health and productivity: productivity, productivity by others, peer relationships, nonwork satisfaction, and stress and irritability.

The survey included several other single-scale estimations of WFH productivity and satisfaction, such as self-reported productivity, satisfaction (with work in general, and with the WFH situation), and happiness. Additionally, participants stated their willingness to continue with WFH. These items were all measured on a 10-point Likert scale, ranging from absolutely not (1) to completely (10). In order to capture the negative spectrum of productivity, a short module measured burnout tendency, comparable to Bloom et al [8]. Adopted from the Maslach burnout inventory [52], 6 questions were scored on a 7-point Likert scale, ranging from never (1) to always (7), capturing emotional exhaustion. In addition to these six items, we added a 7-point Likert scale for sick days as well as break time during office hours.

To assess the physical characteristics of the home office, we included two separate modules. The UC Berkeley Center for the Built Environment (CBE) module assesses the perceived indoor environmental quality [53]. This survey has been extensively used in peer-reviewed research [54, 55] and measures satisfaction with all relevant indoor environmental factors, such as indoor temperature, air quality, lighting, and noise. We also included the physical office characteristics available in the CBE module. These factors focus on satisfaction with a variety of attributes in the (home) office, such as desk, chair, screen, hardware, and Wi-Fi satisfaction. All factors are measured on a 7-point Likert scale, ranging from very dissatisfied (1) to very satisfied (7).

In addition to the CBE module on the environment of and hardware in the home office, we included a set of metrics to further assess indoor environmental quality and a variety of job-related measures. The former included layout of the home office (open versus closed), lighting (natural light versus no natural light), and ventilation (none, mechanical systems like HVAC or fans, or manual methods such as opening windows or doors). Additionally, participants were asked to estimate the surface of their home office (length and width in meters), and how often they ventilated their home office (as a percentage of time spent in the home office). Job-related characteristics included the ability of the respondent to perform their work from home (1–10 scale), the company size (1–5, 5–15, 15–50, 50+ employees), length of the workweek in hours, and job category (e.g., governmental, non-governmental, self-employed, or on-call).

Finally, demographic information included age, gender, income, family size, household situation, and housing characteristics. The household situation could support or hamper productivity as compared to the office situation. The house that respondents reside in could interfere with the perceived quality of WFH office characteristics. We therefore match respondent data, based on 4-digit postcode, to data on average urbanicity ('stedelijkheid'; STED), address-

density ('omgevingsadressendichtheid'; OAD), and house value ('waarde van onroerende zaken'; WOZ).

## Empirical model

**Linear regression models.** We use a simple linear regression (OLS) to formally assess the relationship between home office satisfaction and productivity and burnout measures using the following models:

$$y_i = \alpha_0 + \alpha_1 \text{ hardware satisfaction}_i + \alpha_2 GC_i + \varepsilon_i \tag{1a}$$

$$y_i = \beta_0 + \beta_1 \text{ indoor environment satisfaction}_i + \beta_2 GC_i + \varepsilon_i \tag{1b}$$

where $y_i$ is the predicted value of either productivity or burnout tendency for each participant $i$. Model 1a isolates the effect of home office hardware (hardware satisfaction$_i$), whereas model 1b isolates the effect of the home office environment (indoor environment satisfaction$_i$) on the dependent variables. Both models include a set of carefully selected general controls ($GC_i$) that could otherwise confound the estimators. Specifically, these include demographic characteristics (e.g., gender, age, etc.), job characteristics (company size, job suitable for working from home, type of work, income, and working hours), and household characteristics (household size, children at home, partner at home, and pets).

Model 2 shows the combined model including both the effect of home office hardware and home office indoor environment on our dependent variable $y_i$.

$$y_i = \delta_0 + \delta_1 \text{ hardware satisfaction}_i + \delta_2 \text{ indoor environment satisfaction}_i + \delta_3 GC_i + \delta_4 OC_i + \varepsilon_i \tag{2}$$

This model also adds physical characteristics of the home office as controls ($OC_i$), including lighting, means of ventilation, and the room plan. In the Supporting information, an additional model is shown, in which we match our participants at postcode level to average house characteristics. Running model 2 with and without home office controls, we estimate four models in total for both productivity and burnout tendency.

For all models, we standardized continuous variables, since they are originally measured on different Likert scales, to simplify the interpretation of the coefficients (coefficients are standardized unless specifically mentioned otherwise). As a result, the coefficients are z-scores and must be interpreted such that each coefficient indicates the change in the dependent variable for each standard deviation increase of the independent variable. Upon inspection, S2 Table shows that both desk and chair, as well as screen and the hardware factor, have a correlation ($r$) exceeding 0.70. Since correlations between these variables are not surprising, they can be specified as a combined variable. Thus, for any further analysis, the scores on these two pairs are combined and averaged per participant.

**Structural equation model.** Following the main analysis, we implement a mediation analysis, using structural equation modelling (While our study employs a cross-sectional design, which limits the ability to establish causality compared to a cross-lagged design, extensive research from controlled experimental settings supports the relationship between ventilation and air quality and their effects on satisfaction, well-being, and performance. Nevertheless, we remain cautious and consider our assessment an analysis of direct and indirect associations. For more on this, see the Limitations section). This analysis of direct and indirect associations assesses the impact of the physical environment on productivity, mediated by hardware and indoor environment satisfaction factors. For the analysis, we construct two latent variables, 'Office Hardware' and 'Office Indoor Environment' which each consists of all individual hardware and indoor environment satisfaction variables (see S1 Fig for the loadings per latent

variable). The factors are loaded by the marker variable identification approach. By doing so, the estimators of the latent variables on the dependent variable are fixed on the original 7-point satisfaction. In other words, the estimators indicate the effect per point estimate increase on a 7-point scale identical to the scales of the underlying variables (Following model specification analysis, we find strong covariance between the latent variables 'Office Hardware' and 'Office Indoor Environment', and indicator items desk and chair as well as screen and hardware. Since the correlations between these variables are intuitively not surprising, they can be specified in a saturated model. This saturated model, containing additional parameters estimating those correlations, indeed fits the data better than the restricted model with these correlations fixed to zero (chi-squared difference = 568, DF difference = 3; p < .000; note that we do not combine the pairs desk & chair and screen & hardware pre-analysis in contrast to the multivariate regression, but enter them individually whilst declaring covariance in the SEM model. Doing so increases the Cronbach alpha of both models with 0.05 and improves the overall model fit).

## Results

### Descriptive findings

**Demographics.**   The survey was completed by 1,002 participants of which 58.1% are male, with mean age of 43.89 (SD = 12,54). All participants had work that was at least partially executed from home, with 57.9% of the respondents exclusively working from home. Table 1 shows further demographic characteristics. 54.6% of our sample completed higher education (as compared to just over 40% for the Netherlands more broadly in 2019 [56]) and 53.6% earn more than the median income in the Netherlands. These metrics support the notion that cognitively demanding (desk) jobs are more likely to be suitable to be performed from home [6]. Considering the home office, we find that they are relatively spacious (M = 25.1 m$^2$, SD = 17.4) and predominantly illuminated by natural light (82.6%). Note that we use the estimated length and width of the office (in meters) to calculate the total surface in m$^2$. Extreme values (potential mistakes) for either metric ultimately led to unrealistic outliers. As a result, we truncated the office surface from 2 to 100 m$^2$ (46 data points are excluded).

**Home versus work: Performance differences.**   Table 2 shows the general scoring on the main variables of interest, comparing the home office situation with the office by applying nonparametric Wilcoxon signed-rank tests on paired samples' median differences. For example, the average WFH-HWQ factor productivity score at home is 6.84 out of 10 (SD = 1.28 with a maximum of 9.90). Compared to the office, the WFH-HWQ factor productivity scores higher at work ($p < .001$), whereas self-reported productivity does not differ ($p > .06$). The overall trend for the other WFH-HWQ factors (excluding Stress) shows a higher score for the office. The single-question estimations of productivity and satisfaction show a slightly higher, yet similar, trend. Since S1 Table shows that the WFH-HWQ factor productivity estimator is strongly correlated with its single-question counterpart ($r = .73$, $p < .0001$), we solely refer to the WFH-HWQ productivity factor when we discuss productivity scores.

The average burnout score suggests that most of the respondents show limited signs of burnout while in the home office (on a 7-point scale; M = 2,87, SD = 1,25). This score does not deviate much from similar reports of a larger Dutch sample, which uses the same measurement [57]. Yet, relative to working from home, the office performs better: at home, the burnout tendency is significantly higher compared to the office ($p < .01$).

**Home versus work: Physical differences.**   Fig 2 shows the distribution plots of both the office indoor environmental scores (A-D) and office hardware (E-I) scores. WFH increases the satisfaction with all office indoor environmental factors: Temperature (A), Air Quality (B),

**Table 1. Summary statistics of sample demographics.**

| | | Mean (SD) or N (%) | | | Mean (SD) or N (%) |
|---|---|---|---|---|---|
| **Demographics** | | | **Work Characteristics** | | |
| Age (years) | | 43.9 (12.54) | Income | | |
| Gender (female %) | | 420 (41.9%) | | Modal wage | 184 (18.4%) |
| Education level | | | | Minimum wage | 23 (2.3%) |
| | Low | 65 (6.5%) | | Below modal | 106 (10.6%) |
| | Middle | 390 (38.9%) | | 1-2x modal | 318 (31.7%) |
| | High | 547 (54.6%) | | 2x modal or more | 219 (21.9%) |
| **Family Characteristics** | | | | Don't know/ don't want to say | 152 (15.2%) |
| Household members | | 2.61 (1.21) | Company size (employees) | | |
| Children home during office hours | | | | 1–5 | 101 (10.1%) |
| | No Kids | 482 (48.1%) | | 5–15 | 70 (7.0%) |
| | Always | 33 (3.3%) | | 15–50 | 131 (13.1%) |
| | Sometimes | 333 (33.2%) | | 50+ | 700 (69.9%) |
| | Never | 154 (15.4%) | Work sector | | |
| Partner home during office hours | | | | Governmental | 195 (19.5%) |
| | No Partner | 240 (24.0%) | | Non-governmental | 654 (65.3%) |
| | Always | 234 (23.4%) | | Temp/ on-call worker | 32 (3.2%) |
| | Sometimes | 244 (24.4%) | | Self-employed | 121 (12.1%) |
| | Never | 284 (28.3%) | Contract hours | | |
| Pets | | | | 36+ hours | 610 (60.9%) |
| | Dog | 188 (18.8%) | | 20–35 hours | 303 (30.2%) |
| | Cat | 268 (26.7%) | | 12–19 hours | 49 (4.9%) |
| | | | | Less than 12 hours | 40 (4.0%) |
| **Home Office Characteristics** | | | **Work from Home Characteristics** | | |
| Home office floor plan | | | Working from home currently | | |
| | Open | 377 (37.6%) | | Exclusively from home | 530 (57.9%) |
| | Average | 139 (13.9%) | | Partially from home | 386 (42.1%) |
| | Closed | 486 (48.5%) | | Missing | 86 |
| Home office lighting | | | | Work suitable to perform from home (0–10) | 7.59 (2.39) |
| | Natural | 828 (82.6%) | Work suitable to perform from home (0–10) | | 7.59 (2.39) |
| | Average | 140 (14.0%) | **House Characteristics** | | |
| | No Natural | 34 (3.4%) | Real estate value (WOZ; x €1.000)) | | 274.64 (88.32) |
| Home office ventilation | | | Address density (OAD; per 1 kilometer radius) | | 2118.10 (1749.55) |
| | Mechanic | 135 (13.5%) | Urbanity (STED; Categorical address-density) | | |
| | Manual | 825 (82.3%) | | Extremely High Urbanity | 199 (25.8%) |
| | None | 42 (4.2%) | | High Urbanity | 248 (32.2%) |
| Home office surface ($m^2$) | | 25.14 (17.40) | | Average Urbanity | 154 (20.0%) |
| | | | | Low Urbanity | 104 (13.5%) |
| | | | | Non-Urban | 66 (8.6%) |
| | | | | Missing | 231 |

**Table 2. Satisfaction and productivity: Home office versus the office.**

| | Home Office (N = 1,002) | The Office (N = 1,002) | *p*-value |
|---|---|---|---|
| **Office Indoor Environment Satisfaction** *(scale: 1–7)* | | | |
| Temperature | 5.13 (1.28) | 4.59 (1.24) | 0.00*** |
| Air quality | 5.41 (1.12) | 4.61 (1.27) | 0.00*** |
| Lighting | 5.37 (1.20) | 5.07 (1.30) | 0.00*** |
| Noise | 5.36 (1.32) | 4.63 (1.34) | 0.00*** |
| **Office Hardware Satisfaction** *(scale: 1–7)* | | | |
| Desk | 4.41 (1.55) | 5.46 (1.12) | 0.00*** |
| Chair | 4.50 (1.58) | 5.37 (1.14) | 0.00*** |
| Screen | 4.86 (1.53) | 5.52 (1.09) | 0.00*** |
| Hardware | 5.19 (1.32) | 5.44 (1.07) | 0.00** |
| WiFi | 5.23 (1.32) | 5.52 (1.15) | 0.00*** |
| | Home Office (N = 1,002) | The Office (N = 1,002) | *p*-value |
| **WFH-HWQ Factor Scores** *(scale: 1–10)* | | | |
| Productivity | 6.84 (1.28) | 7.11 (0.93) | 0.00*** |
| Productivity by others | 7.55 (1.24) | 7.78 (1.04) | 0.01* |
| Stress and irritability | 3.82 (1.63) | 3.95 (1.55) | 1.00 |
| Peer relations | 6.65 (1.59) | 7.41 (1.23) | 0.00*** |
| Non-work satisfaction | 5.99 (1.59) | 7.59 (1.06) | 0.00*** |
| **Single-Item Scale Scores** *(scale: 1–10)* | | | |
| Satisfaction work situation | 6.82 (1.90) | 7.22 (1.62) | 0.00*** |
| Happy with work situation | 6.76 (1.96) | 7.30 (1.56) | 0.00*** |
| Self-reported productivity | 7.16 (1.72) | 7.47 (1.35) | 0.06 |
| **Burnout Tendency** *(scale: 1–7)* | | | |
| Burnout metric | 2.87 (1.25) | 2.64 (1.10) | 0.00*** |

Note.

*$p<0.1$,

**$p<0.05$,

***$p<0.01$.

Lighting (C), and Noise (D) all score higher as compared to the work environment (mean scores range between 5.37 and 5.13 for the home office, compared to 5.07 and 4.59 for the office; on a 7-point Likert scale). For office hardware, we observe the opposite trend: overall office hardware satisfaction is higher in the office. The satisfaction for the desk (E), chair (F), screen (G), hardware (H), and Wi-Fi (I) range between 5.23 and 4.41 at home, whereas the office hardware satisfaction levels range between 5.52 and 5.37. Table 2 shows that all differences are statistically significant, using the non-parametric Wilcoxon rank sum test and Bonferroni multiple comparison corrections. These observations support the notion that at home, optimizing ergonomics (e.g office hardware factors) remains challenging [40] while increased individual control over office indoor environment is preferred [42].

It is important to confirm that respondents are considering and rating their home office as distinctly different from their office. We correlate each variable's score at home and at the office. As shown in S2 Table, scores correlate moderately with different variables within the same environment (home office or regular office), but correlations are much lower between the same variables in different environments. For instance, the correlation between

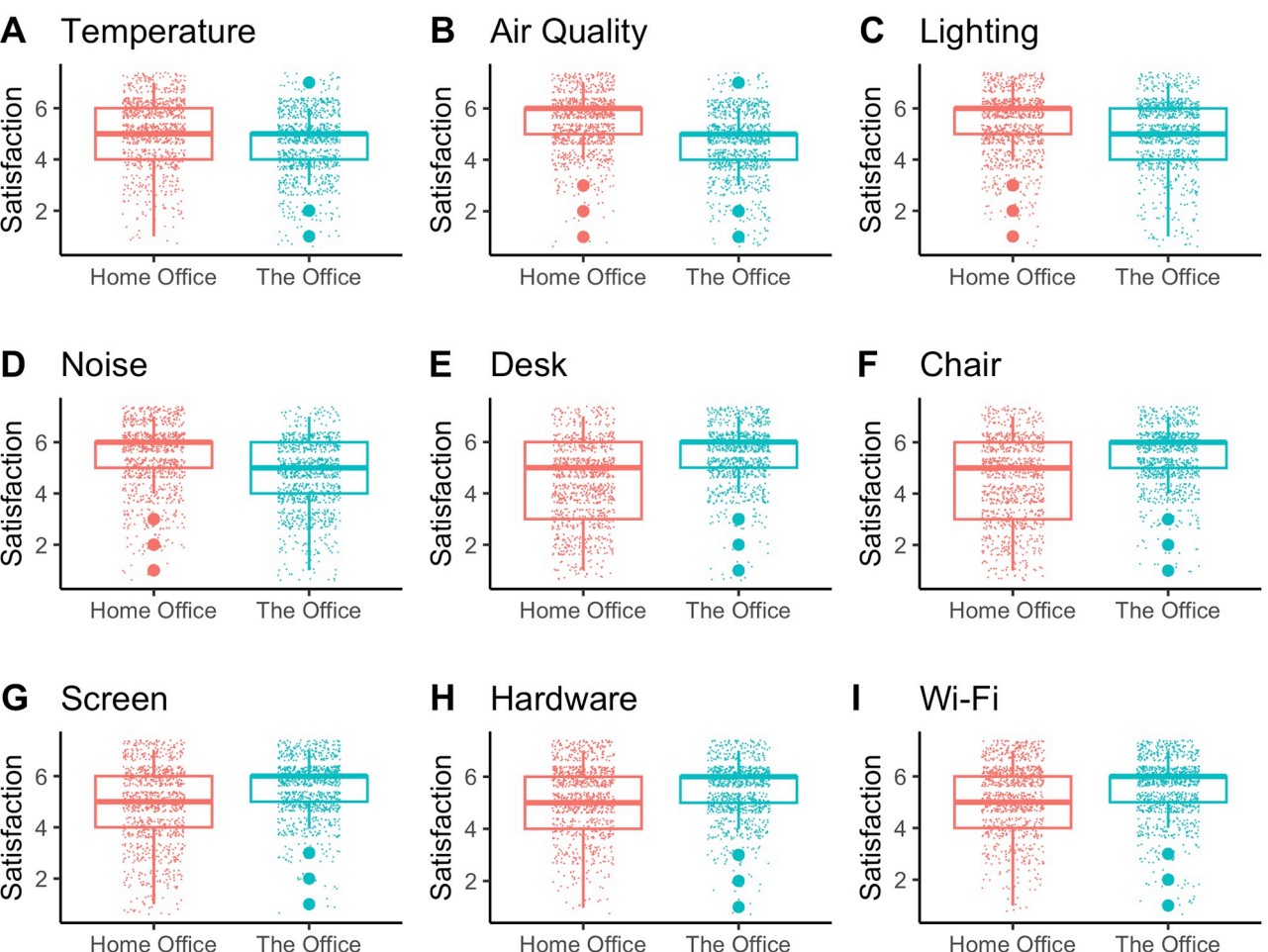

**Fig 2. Hardware and indoor environment: Home office versus the office.** Comparison of satisfaction levels between home and the office environments across various factors. Panels A-D show indoor environment satisfaction ratings for (A) Temperature, (B) Air Quality, (C) Lighting, and (D) Noise, whereas Panels E-I show hardware satisfaction ratings for (E) Desk, (F) Chair, (G) Screen, (H) Hardware, and (I) Wi-Fi. Each plot includes boxplots and data distributions, with orange representing home office and blue representing traditional office settings.

temperature and noise at home is $r = 0.41$, which is considered a moderately strong correlation. Comparatively, the correlation between the temperature at the office and the home office is negligible ($r = 0.06$).

## Regression results

**Explaining productivity and burnout in the home office.** Table 3 shows the estimated standardized coefficients and standard errors of the home office hardware and home office indoor environment variables in explaining productivity. Models 1–4 show that all office hardware variables at home are positively associated with productivity, such that increased satisfaction with each office hardware variable is associated with an increase in productivity when WFH (coefficients ranging from 0.18 to 0.15; SD = .03 to .05). For example, a 1.32 increase of Wi-Fi satisfaction on a 0–7 satisfaction scale translates to a 0.23 increase on a 0–10 productivity scale. This effect is relatively strong, comparable to the effect of, for example, sometimes having children at home during working hours to having no children at home (see S3 Table).

**Table 3. Regressions of office hardware and indoor environment satisfaction on productivity and burnout tendency.**

| | Productivity | | | | Burnout Tendency | | | |
|---|---|---|---|---|---|---|---|---|
| | (1) | (2) | (3) | (4) | (5) | (6) | (7) | (8) |
| Desk & Chair | .15 (.04)*** | | .09 (.04)** | .09 (.04)** | −.14 (.04)*** | | −.10 (.04)** | −.11 (.05)** |
| Screen & Hardware | .18 (.05)*** | | .11 (.05)** | .11 (.05)** | − .09 (.05)* | | −.04 (.05) | −.03 (.05) |
| WiFi | .18 (.03)*** | | .10 (.04)*** | .10 (.04)*** | −.12 (.04)*** | | −.07 (.04)* | −.07 (.04)* |
| Temperature | | .14 (.04)*** | .09 (.05)** | 10 (.05)** | | −.07 (.04)* | −.04 (.04) | −.06 (.04) |
| Air Quality | | .08 (.04)* | .03 (.04) | .04 (.04) | | −.11 (.04)** | −.08 (.04)* | −.09 (.04)** |
| Lighting | | .11 (.04)*** | .07 (.04)* | .06 (.04) | | −.05 (.04) | −.02 (.04) | .01 (.04) |
| Noise | | .21 (.04)*** | .16 (.04)*** | .16 (.04)*** | | −.13 (.04)*** | −.10 (.04)*** | −.09 (.04)** |
| General Controls | Yes | Yes | Yes | Yes | Yes | Yes | Yes | Yes |
| Home Office Controls | No | No | No | Yes | No | No | No | Yes |
| Observations | 1,002 | 1,002 | 1,002 | 956 | 1,002 | 1,002 | 1,002 | 956 |
| $R^2$ | .25 | .27 | .30 | .30 | .18 | .19 | .21 | .21 |
| Adjusted $R^2$ | .23 | .25 | .28 | .27 | .16 | .17 | .18 | .17 |
| Residual Std. Error | .88 (df = 972) | .87 (df = 971) | .85 (df = 968) | .85 (df = 915) | .92 (df = 972) | .91 (df = 971) | .91 (df = 968) | .90 (df = 915) |
| F Statistic | 11.41*** (df = 29; 972) | 12.16*** (df = 30; 971) | 12.79*** (df = 33; 968) | 10.01*** (df = 40; 915) | 7.60*** (df = 29; 972) | 7.71*** (df = 30; 971) | 7.70*** (df = 33; 968) | 6.06*** (df = 40; 915) |

Note.

*$p<0.1$,

**$p<0.05$,

***$p<0.01$.

Standard error in parentheses. Models 1 to 4 regress on Productivity, whereas models 5 to 8 regress on burnout propensity. For the full model including all controls, see S3 Table (productivity) and S4 Table (Burnout Tendency).

The home office indoor environment variables show a similar pattern: without exception, all variables are associated with increased productivity (coefficients ranging from 0.21 to 0.08; SD = .04). Combining both home office hardware and indoor environment variables in model 3 decreases the size of the coefficients for some variables in the productivity model. Adding additional controls in model 4 hardly affects the model: all office hardware variables remain relevant predictors of productivity, as well as temperature and noise satisfaction (indoor environment).

Table 3, models 5–8, show the coefficients for the same home office hardware and home office indoor environment satisfaction on burnout tendency. For the burnout models, the association is negative, meaning that an increase in satisfaction on either variable's satisfaction is associated with a decrease in the individual level of feeling burnout. The most robust predictors of burnout tendency are desk, chair and Wi-Fi satisfaction (home office hardware), as well as air and noise satisfaction (home office indoor environment).

Comparing both tables shows that, on average, office hardware and indoor environment coefficients and significance levels are generally higher in the productivity models. For example, noise satisfaction is meaningful for both productivity as well as burnout tendency, yet the coefficient is about 50% higher for productivity in all models (0.21 to 0.16 versus -0.13 to -0.09, for productivity and burnout, respectively).

**Individual heterogeneity.** Factors other than hardware and indoor environment, for example, household characteristics, may also affect productivity and burnout Tendency. S3 and S4 Tables report the full specifications of Model 3. The results show that the degree to which work can be performed from home does not add predictive value to our model. Women

tend to report higher levels of productivity ($\delta = 0.15$, SD = .07). Not living alone, i.e., having a larger household, decreases burnout score and increases productivity ($\delta = -0.10$, SD = .04; $\delta = 0.11$, SD = .04, respectively). Having a partner who is not (or only sometimes) home during office hours is associated with increased productivity ($\delta = 0.14$–0.15, SD = .08) compared to the baseline of having no partner at all. In that sense, having a partner seems good for productivity, if they are not constantly present at home during working hours. For children, a predictable, strong, and linear relationship emerges: burnout tendency increases and productivity decreases when children spend more time at home during working hours. Interestingly, having a dog increases the burnout score significantly ($\delta = 0.17$, SD = .08).

Finally, previous research indicated that during the pandemic, young employees seemed to appreciate WFH more, and opted for the home office more often as compared to older employees [28]. Contrasting, we find that the difference between older and younger respondents is negative: the difference between 20-years old versus to 40-years old is an increase in the WFH productivity score of about 0.25 (on a scale from 1–10). In terms of economic significance, this effect is twice as strong as the gender effect on productivity. In addition, we document that older respondents report a stronger willingness to continue to WFH (0.01 standard deviation increase per year of age, SD = .003; see S5 Table). Together, our results reflect that older workers not only report to be more productive at home and at the office than younger workers, but also seem to have an overall higher willingness to continue to WFH.

**Mediation analysis.** We extend our analysis by exploring whether behavior at home (as it relates to using the home office) is associated with satisfaction with hardware and indoor environment. Although office characteristics are fixed or dependent on capital expenditures, the indoor environment can to a large extent be manipulated by human actions. Specifically, we measure the behavior of respondents working from home through active ventilation, both at the extensive and intensive margin.

We implement a mediation analysis through structural equation modelling in order to understand how the home office environment is associated with productivity. Our model specifications show that the 'Office Hardware' and 'Office Indoor Environment' item loadings are meaningful per latent factor. Further reliability calculations confirm the factor's consistency, with both factors showing a Cronbach alfa above 0.8 (a = 0.80 and a = 0.85, for 'Office Hardware' and 'Office Indoor Environment', respectively). Additional model fit tests confirm that our saturated model fits the data well (CFI/TLI > .95, RMSEA close to .05, and SRMR < .05).

First, the latent variables 'Office Hardware' and 'Office Indoor Environment' have a strong and distinct direct association with WFH productivity, as can be seen in Fig 3. For both factors, a standard deviation increase is associated with around a 0.3 standard deviation increase in

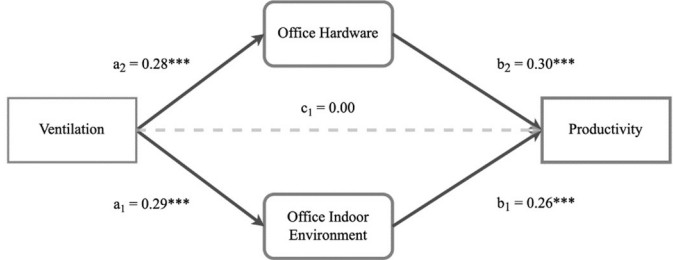

**Fig 3. Structural equation model: Productivity.** Structural equation model depicting the relationships between ventilation, office hardware, office indoor environment, and productivity. Paths are labelled with standardized regression coefficients. Solid lines indicate significant relationships, while the dashed line indicates a non-significant relationship. All coefficients of solid lines are significant at ***p < 0.001.

productivity. Second, the percentage of time that the home office is ventilated is significantly associated with both increased hardware and indoor environment satisfaction. Each standard deviation increase in ventilation of the office increases satisfaction with 0.29 and 0.27 points, respectively. Third, ventilation no longer shows a direct association with productivity, which is not captured by its relation to hardware or indoor environment satisfaction (p = 0.88). Hence, the association of ventilation with productivity is fully mediated by satisfaction with hardware or the indoor environment. Both indirect unstandardized parameters via the latent variables are estimated at 0.002, with a total estimated effect of ventilation on productivity of 0.004. Thus, moving from 0% to 100% ventilation of the office is associated with a productivity increase of 0.4 on the 10-point scale through higher hardware or indoor environment satisfaction. Considering that the average productivity score is 6,11 (SD = 1,06), the magnitude of this association is not trivial. This effect equates to 8.18% of the mean and 47% of the standard deviation of the productivity variation in our sample.

Replacing productivity with burnout tendency or willingness to continue WFH in the model shows the same mediation association. Both models, shown in Fig 4, are well-fitted (both show CFI/TLI > .95, RMSEA close to .05, and SRMR < .05), and for both models, the association runs fully through the latent variables. The total estimated effect of ventilation on burnout tendency is -0.004, with comparable mediation through satisfaction with home office hardware and environment. Moving from 0% to 100% ventilation of the home office is associated with a burnout tendency decrease of 0.4 on the 7-point scale. For the willingness to continue with WFH, the significance and strength of association are stronger for hardware compared as compared to the indoor environment (a = 0.003, p = 0.016; a = 0.005, p < .000, respectively). Moving from 0% to 100% ventilation of the office is associated with an increased willingness to continue WFH of 1.2 on the 10-point scale.

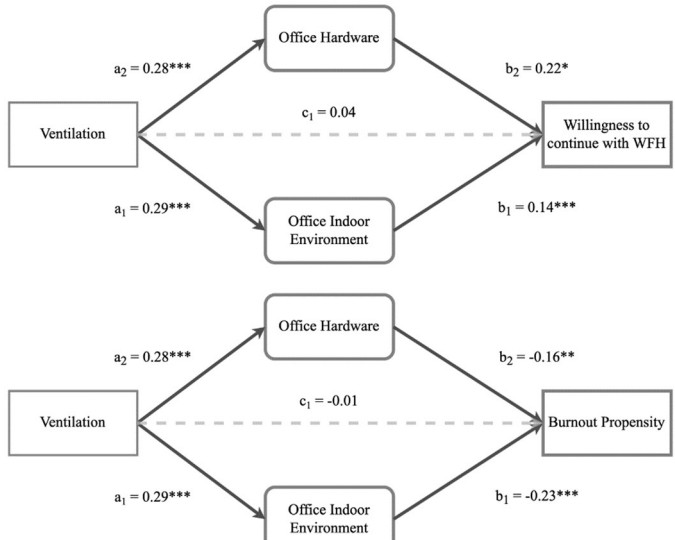

**Fig 4. Structural equation model: Burnout tendency and willingness to WFH.** Structural equation model depicting the relationships between ventilation, office hardware, office indoor environment, and willingness to continue with WFH (top) or Burnout propensity (bottom). Paths are labelled with standardized regression coefficients. Solid lines indicate significant relationships, while the dashed line indicates a non-significant relationship. Solid lines coefficients are significant at $^*p < 0.05$, $^{**}p < 0.01$, and $^{***}p < 0.001$.

## Discussion and conclusion

### Discussion

The success of WFH, and the likelihood of its continuation after the pandemic, is dependent on sustained employee satisfaction with and employee productivity in the home office environment. But satisfaction and productivity, in turn, may also be influenced by the physical characteristics of the home office. We use survey data to study the effect of home office satisfaction and environment-improving behavior on productivity, burnout, and willingness to continue with WFH.

Comparing WFH with working from the office first shows that the self-reported productivity is lower at home compared to working at the office. This is contrasting earlier findings based on self-reported productivity, but consistent with multiple non-self-reported outcome analysis [6, 15, 16]. When looking at the physical characteristics of the office, we find that the indoor environmental satisfaction appears higher at home, whereas physical hardware satisfaction such as desks and chairs are preferred at the office. This implies that optimizing ergonomics at home remains challenging [40] while individually being in control of the indoor environment at home is preferred [42]. Overall, we find a relatively low score for the willingness to continue WFH, in contradiction to many recent reports, which supports a deeper investigation into factors facilitating successful WFH [4, 58].

The association between the both home office hardware as well as indoor environment satisfaction and productivity is profound. Higher satisfaction in both these domains is associated with higher WFH productivity and lower burnout tendency. The majority of all indoor environment and hardware factors included in this paper (with the exception of air quality) are associated with increased productivity and decreased burnout tendency. We find heterogeneity in the reported effects–women and larger households seem to be more productive at home, while having children at home decreases productivity and increases burnout scores. Having a partner increases productivity, but only when they are not around during office hours. Finally, we find that older workers report being more productive, having lower burnout scores, and stating to be more willing to continue to WFH compared to younger workers, contrasting existing evidence [28, 29].

To show the influence that real behavior could have on WFH success, we investigate the association of ventilation with productivity. By means of a mediation analysis, we confirm that the amount of time that the home office is ventilated is not only directly associated with increased satisfaction but also indirectly with increased productivity. Practically, we find that changing from not ventilating to ventilating the home office all the time (moving from 0% to 100%) is indirectly associated with 0.5 points on the 10-point scale increased productivity. The magnitude of this estimate on productivity is comparable to moving from no children at home to always having children at home during working hours (0.7-point decrease of productivity). In addition, moving from 0% to 100% ventilating time is associated with 0.4 points on a 7-point scale decreased burnout tendency, and 1.2 points on a 10-point scale increased willingness to continue with WFH. Hence, we find that ventilating the home office is a crucial underlying factor predicting overall satisfaction and is indirectly associated with increased productivity, increased willingness to WFH, and decreased burnout tendency.

### Implications

The main contribution of this paper is to show that the physical characteristics of the home office, including the indoor climate, is associated with employee productivity and satisfaction when WFH. Specifically, we not only connect the outcomes of WFH to self-reported

satisfaction, but also to behavior that actively influences the indoor environmental quality. The move from the office to the home office needs to be combined with careful design and investment in the quality of the office *and* its indoor climate. Failure to do so is not only likely to be associated with decreased productivity, but also decreased willingness to work from home, and increased burnout tendency. The physical climate is a determining factor in successful work from home prolongation. As such, this paper reaffirms that the effect of a healthy indoor climate affects productivity, related to previous research that shows significant health effects of indoor climate [33, 34, 59, 60].

Additionally, our results also suggest that it is crucial to objectively measure the quality of the physical environment, as merely collecting self-reported satisfaction scores might paint an incomplete or even incorrect picture. This is not only shown by the fact that satisfaction scores are influenced by improved ventilation, but also by the fact that self-reported air quality satisfaction, the closest subjective measure related to ventilation, is not associated with productivity. Thus, solely based on self-report analysis, ventilation would have been an unlikely factor considered to improve the success of WFH. Since evaluations of working generally, as well as evaluations of indoor air quality specifically, are heavily reliant on self-reported scores, this conclusion is not trivial.

## Limitations

Our results have some limitations. Self-reported data may introduce common method variance, potentially affecting the relationships between predicting, mediating, and outcome variables [61]. We counter these effects by deemphasizing the compartmentalization of work conditions and characteristics with productivity (outcome) measures. Although we do not measure objective productivity, we at least partially alleviate this concern by using an extensively validated questionnaire.

Second, practical constraints limited our ability to implement a cross-lagged design with multiple measurement points which is considered the normative approach for establishing temporal precedence and causality in mediation analysis [62, 63]. Our study employed a cross-sectional design, which has precedent in the literature [64, 65], and can still provide valuable insights. We justify our approach based on the model fit and both theoretical as well as literature support of the causal role of air quality improvement on satisfaction and performance [36, 41, 66]. However, we acknowledge that this approach does not allow for the determination of causality with the same rigor as a longitudinal design. Future research should aim to utilize cross-lagged designs to further validate our findings and establish clearer causal relationships.

Third, data on in-office work comes from recall data which may be biased [67] or influenced by the broader undesirability of pandemic-era work [68, 69]. We report on differences between the situation during and before COVID-19 (at the office). To do so, we did not ask our participants at that time, but rather asked them to recollect from memory. Unfortunately, recollection itself is less accurate than asking in the current situation [67, 69]. The current situation could even influence the recollected score, as it serves as a reference point [51, 68]. The mere fact that WFH is mandatory could put the productivity at work (as well as life in general) in a more generous daylight that it truly was. Taken together, our data quality would have improved if we had foreseen the pandemic, and pretested our subject before the outbreak. Alas.

Finally, the extraordinary circumstances surrounding the COVID-19 pandemic itself could be reflected in our subjective scores, making the observed behaviors, attitudes, and outcomes different from remote work under more typical conditions (mood-as-information theory [70].

While our research provides valuable insights into the pandemic WFH experience, caution should be exercised when generalizing these findings to other contexts or periods.

## Conclusion

In conclusion, we find strong evidence that a favorable home office is associated with multiple WFH success outcomes. Moreover, air-quality-improving behavior is associated with home office satisfaction improvements. The move from the work office to the home office needs to be combined with intentional design and investment in the quality of the office *and* its climate. Failure to do so will likely have adverse ramifications for the future of WFH [5, 71, 72].

## Supporting information

**S1 Table. Correlation table: Productivity and stress.**
(DOCX)

**S2 Table. Correlation table: Hardware and indoor environment.**
(DOCX)

**S3 Table. Regression results: Full specification (Productivity).**
(DOCX)

**S4 Table. Regression results: Full specification (Burnout tendency).**
(DOCX)

**S5 Table. Regression results: Full specification (Willingness to continue WFH).**
(DOCX)

**S1 Fig. Structural equation model latent variables loading and covariance.**
(TIF)

## Author Contributions

**Conceptualization:** Martijn Stroom, Piet Eichholtz, Nils Kok.

**Data curation:** Martijn Stroom, Nils Kok.

**Formal analysis:** Martijn Stroom.

**Methodology:** Nils Kok.

**Supervision:** Nils Kok.

**Writing – original draft:** Martijn Stroom.

**Writing – review & editing:** Piet Eichholtz, Nils Kok.

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
