## [Decision Letter · Decision Letter 0]

13 Sep 2023

PONE-D-23-05772Does working from home work? That depends on the home.PLOS ONE

Dear Dr. Nils Kok,

Thank you for submitting your manuscript to PLOS ONE. After careful consideration, we feel that it has merit but does not fully meet PLOS ONE’s publication criteria as it currently stands. Therefore, we invite you to submit a revised version of the manuscript that addresses the points raised during the review process.

ACADEMIC EDITOR: We have now received two reviewers’ reports for your manuscript. In addition to some other issues, reviewers asked several queries regarding the validity of the scale, and language ambiguity, which are all valid concerns. If you are not using panel or longitudinal data, words such "causality or impact or effect" should be removed. The reviewer’s points are clearly laid out in their reviews which are appended below.

We look forward to receiving your revised manuscript.

Kind regards,

Md Mohsan Khudri

Academic Editor

PLOS ONE

Journal Requirements:

4. Please upload a new copy of Figures 1 and 2 as the detail is not clear. Please follow the link for more information: https://blogs.plos.org/plos/2019/06/looking-good-tips-for-creating-your-plos-figures-graphics/" https://blogs.plos.org/plos/2019/06/looking-good-tips-for-creating-your-plos-figures-graphics/

Reviewers' comments:

Reviewer's Responses to Questions

**Comments to the Author**

1. Is the manuscript technically sound, and do the data support the conclusions?

Reviewer #1: Yes

2. Has the statistical analysis been performed appropriately and rigorously? 

Reviewer #1: Yes

3. Have the authors made all data underlying the findings in their manuscript fully available?

Reviewer #1: Yes

4. Is the manuscript presented in an intelligible fashion and written in standard English?

Reviewer #1: Yes

5. Review Comments to the Author

Reviewer #1: This contribution explores the role of indoor environment, office hardware and environmental factors (ventilation) related to working from home (WFH) for employees’ productivity, burnout, and motivation to continue WFH. Results are based on survey data of a Dutch panel (Flycatcher) of employed people with WFH experience (N=1002). Main findings support the importance of an agreeable physical climate, mediated by office hardware and indoor environment, for well-being at, and preference for, WFH.

In terms of methods and statistical analysis, the manuscript offers good quality. The same holds true for integration of current state of research. Moreover, the quality of English writing is appropriate. My main criticism concerns the way this manuscript is organised as well as its extensive length. In more detail, major concerns are as follows:

1. There is no justification of authors’ deviation from the usual structure of scientific papers (Introduction, Methods, Results, Discussion, Conclusions). In a rather strange way, authors mix up introductory parts with methods (p. 8ff of MS), and they mix up presentation of results with discussion (p. 23f.), while failing to present a separate Discussion section including strengths and limitations of own study.

2. The manuscript is definitely too lengthy in its current version. Readers expect a more concise presentation of essential information. The whole part from p. 8 to p. 20 requires substantial cuts.

3. While authors put a lot of effort into methodological details, they do not reflect on limitations of their study, such as limited validity of self-reported data (e.g. common method variance between ‘predicting', ‘mediating’ and ‘outcome’ variables) or limited generalisation of findings (e.g. sample representativeness; COVID-19 specific effects). Authors use a biased language when mentioning ‘causal indicators’ (p. 25) or ‘burnout proneness’ (p. 22 and passim), suggesting availability of longitudinal data.

6. PLOS authors have the option to publish the peer review history of their article (what does this mean?). If published, this will include your full peer review and any attached files.

Reviewer #1: **Yes: **Johannes Siegrist

---

## [Decision Letter · Decision Letter 1]

24 May 2024

PONE-D-23-05772R1Does working from home work? That depends on the home.PLOS ONE

Dear Dr. Kok,

Thank you for submitting your manuscript to PLOS ONE. After careful consideration, we feel that it has merit but does not fully meet PLOS ONE’s publication criteria as it currently stands. Therefore, we invite you to submit a revised version of the manuscript that addresses the points raised during the review process.

We look forward to receiving your revised manuscript.

Kind regards,

Daphne Nicolitsas

Academic Editor

PLOS ONE

Journal Requirements:

Reviewers' comments:

Reviewer's Responses to Questions

**Comments to the Author**

1. If the authors have adequately addressed your comments raised in a previous round of review and you feel that this manuscript is now acceptable for publication, you may indicate that here to bypass the “Comments to the Author” section, enter your conflict of interest statement in the “Confidential to Editor” section, and submit your "Accept" recommendation.

Reviewer #2: (No Response)

Reviewer #3: (No Response)

2. Is the manuscript technically sound, and do the data support the conclusions?

Reviewer #2: Yes

Reviewer #3: Partly

3. Has the statistical analysis been performed appropriately and rigorously? 

Reviewer #2: Yes

Reviewer #3: Yes

4. Have the authors made all data underlying the findings in their manuscript fully available?

Reviewer #2: Yes

Reviewer #3: Yes

5. Is the manuscript presented in an intelligible fashion and written in standard English?

Reviewer #2: No

Reviewer #3: Yes

6. Review Comments to the Author

Reviewer #2: Thank you for the revisions. The manuscript reads significantly better.

Couple minor edits:

Note that in Table 1., the value for high education level is missing.

Line 208: It is unclear what is meant by manual of mechanical ventilation, and 0% to 100%. Is "manual" the ability to change the air flow or fan, or is it simply opening the window? Or do the authors actually mean air quality? Or perhaps the authors mean 'no' ventilation vs. 'having' manual or mechanical ventilation. Please clarify in the main text.

Reviewer #3: I find your manuscript very interesting and timely. Having data supported information about the physical environment and its relationship to productivity is really crucial in the era of changing work environment. However, you use correlational strategy, therefore, you should completely avoid using causal language in the manuscript. Moreover, to make mediation results sound, you have to use cross-lagged design with at least two measurement point.

7. PLOS authors have the option to publish the peer review history of their article (what does this mean?). If published, this will include your full peer review and any attached files.

Reviewer #2: No

Reviewer #3: No

---

## [Editor Report · Decision Letter 2]

18 Jun 2024

Does working from home work? That depends on the home.

PONE-D-23-05772R2

Dear Dr. Kok,

We’re pleased to inform you that your manuscript has been judged scientifically suitable for publication and will be formally accepted for publication once it meets all outstanding technical requirements.

Kind regards,

Daphne Nicolitsas

Academic Editor

PLOS ONE
---

## [Editor Report · Acceptance letter]

16 Jul 2024

PONE-D-23-05772R2 

PLOS ONE

Dear Dr. Kok, 

I'm pleased to inform you that your manuscript has been deemed suitable for publication in PLOS ONE. Congratulations! Your manuscript is now being handed over to our production team.

Kind regards, 

on behalf of

Dr. Daphne Nicolitsas 

Academic Editor

PLOS ONE